# Study on Influence Mechanism of Short-Cut BF Dispersion Morphological Behavior on Concrete Properties Based on Meso Scale

**DOI:** 10.3390/ma15082788

**Published:** 2022-04-11

**Authors:** Zi-Lu Liu, Ye Li, Xin-Ming Chen, Hua-Zhe Jiao

**Affiliations:** 1State Key Laboratory for Geomechanics and Deep Underground Engineering, School of Mechanics and Civil Engineering, China University of Mining and Technology, Xuzhou 221116, China; liuzl@cumt.edu.cn; 2School of Civil Engineering, Henan Polytechnic University, Jiaozuo 454000, China; hpucxm@sina.com (X.-M.C.); 18118419090@163.com (H.-Z.J.)

**Keywords:** basalt fiber concrete, mechanical property, morphological characteristics, pore structure, dispersion

## Abstract

The orientation, distribution, and contact point density of BF (basalt fiber) in the concrete matrix play significant roles in the mechanical properties of BF concrete, but represent a weak point in current research. It is meaningful to study the morphological characteristics of BF in concrete. In this study, the transparent model test and joint blocking method were innovatively adopted to investigate the correlation of dosage with the BF morphological parameters and concrete mechanical properties. A focus on a BF dosage of 0–7.5 kg/m^3^ and the contribution index of fibers C_f_ was defined. Furthermore, NMR and CT techniques were used to observe the changes in the microstructure of BF concrete. The experimental results show that the BF contribution index C_f_ reaches the largest value when the BF content is around 3 kg/m^3^, approximately 2.7; in this case, the mechanical properties of BF concrete were also optimal, and the C_f_ was only 2.34 when the BF content was 7.5 kg/m^3^. NMR and CT test results show that there is a strong correlation between the BF morphological parameters and the distribution of pore structure in the concrete matrix. The overlapping contact of BF clusters led to the penetration of pores, which led the macro-pore proportion to increase dramatically. The increase in the macro-pore proportion is the main reason for the deterioration in concrete performance. In addition, these macro-pores may have adverse effects on the chloride ion permeability of BF concrete.

## 1. Introduction

Fiber-reinforced concrete has been widely applied in civil infrastructure construction and underground engineering in recent years [1,2]. Among various types of fibers, steel fiber has an excellent elastic modulus and tensile strength, and adding steel fibers to concrete can improve the mechanical properties of concrete. However, steel fiber can easily agglomerate, causing a reduction in the pumpability of the concrete [3]. Carbon fiber also has excellent mechanical properties, but the high cost limits its application [4]. Basalt fiber (BF) is a new type of silicate fiber. Because of its natural compatibility, superior mechanical properties, good chemical stability, and affordability (around USD 1.1 per kilogram), it has become a research focus in recent years [5,6]. It is agreed that adding an appropriate amount of BF into concrete can improve the performance of the concrete. However, numerous experiments have shown that the gain effect will decrease if BF is added to concrete excessively, even leading to the deterioration of the concrete. Dilbas et al. [7] found that BF can improve the fracture toughness and compressive strength of treated recycled aggregate concrete (TRAC) when the BF volume fraction is less than 1%, but the compressive and fracture toughness of TRAC are even lower than those of plain TRAC when BF of more than 1.3% is used. Barabanshchikov et al. [8] found that the tensile and flexural strength of fly ash concrete are the best when the BF volume fraction is more than 1% and less than 1.3%. When the BF volume fraction is more than 1.3%, the tensile and flexural strength of HSFAC decrease gradually. This means that there is an optimal value of BF content in concrete, and the optimal BF content may be different for different types of concrete. There are scholars who suggest that this phenomenon is mainly caused by the agglomeration of BF, which is similar to steel fiber. However, this does not fully explain the reasons for the deterioration of concrete.

As a composite material, concrete’s mechanical performance is directly affected by its meso-structure [9,10,11]. In the literature, the related studies about the influence of BF content on the concrete mechanical properties are mainly based on the macro-scale or micro-scale, and they rarely focus on the morphological characteristics and effect mechanism of basalt fiber based on the meso-scale. The meso-scale of BF-reinforced concrete mainly involves BF morphological characteristics and pore structure. The morphological parameters (orientation, distribution, contact point density, etc.) of BF have a strong influence on the concrete meso-structure; they not only can affect the gain effect of BF, but also change the pore structure of concrete, which is probably the main reason for the deterioration in the mechanical properties of concrete. Therefore, it is of great significance to further optimize the BF-reinforced concrete’s performance to explore the morphological characteristics of BF in the concrete matrix.

However, due to the opaque features of concrete, it is challenging to directly observe its inner meso-structure and the actual distribution of fibers; thus, it is necessary to adopt some advanced detection methods [12], such as X-ray scanning, fluorescent tracking, MIP, NMR, CT, etc. The X-ray scanning method seeks to extract the distribution of fibers in the concrete matrix according to the difference in materials in the absorption capacity of X-rays. Olubisi et al. [13] studied the correlation between steel fiber distribution and concrete toughness by the X-ray method; the distribution of steel fiber in concrete was obtained, and it was found that the uniformity of steel fiber in concrete has a significant influence on the toughness of concrete. When the steel fiber is deposited in the middle and bottom of the sample, the toughness of sample decreases dramatically. The fluorescent tracking method involves adding dyed fibers into the concrete, and then capturing the fluorescence signal by using a tracking instrument and CCD camera; in this way, the distribution form of the fibers can be obtained. Lee, B.Y. et al. [14,15] used a fluorescence spectrometer, fluorescence microscope, and image analysis software to obtain the fluorescence characteristics of PVA fiber and analyzed its fluorescence image in FRCC; the dispersion coefficient of PVA fiber was defined and calculated, providing a reference for the utilization efficient of PVA fiber in concrete. However, all these methods have some limitations in certain aspects. X-ray and CT methods are mostly applicable to fibers with a large difference from concrete density, such as steel fibers and carbon fibers. As the BF density and absorption capacity of X-rays are close to that of concrete, the X-ray and CT methods are not suitable for the study of basalt fiber morphology; nonetheless, the CT method is a powerful tool to study the pore structure of concrete. Fluorescent tracking methods are mostly applicable to organic fibers, and require strict test conditions, as factors such as temperature and pH value will affect the accuracy of the test results [16,17]. For BF (non-organic fiber), fluorescent tracking methods are also not applicable. Xue et al. [18] studied the microstructure characteristics of cement mortar by using NMR and MIP technology, and found that curve temperature is a significant factor in meso-structures of concrete, and explained the relationship between meso-pore structure and strength of concrete. NMR and MIP technology are powerful tools to study the meso-structure of concrete, but NMR and MIP are not suitable for the study of fiber morphology. Since the main component of BF is silicate, single detection means it cannot obtain the BF dispersion and concrete meso-structure information simultaneously. Thus, it is necessary to combine multiple methods to study the sample from different scales.

At present, there are still barely suitable methods to study the dispersion form of BF (silicate composition) in concrete. To clarify the dispersion characteristics of BF and the influence mechanism of the morphological characteristics on the meso-structure, in this study, a transparent model test was adopted innovatively, which is frequently used in the field of geotechnical engineering. The distribution of BF in concrete was simulated by a transparent model, which was composed of quartz sand and CaBr_2_ solution, and the joint blocking method was used to analyze the morphological parameters (orientation, distribution) of BF. Compared with previous methods, the transparent model provides a new way to clarify BF dispersion characteristics. In addition, we combined this with NMR and CT technology to study the meso-structure of BF-reinforced concrete. The NMR technique was used to obtain the inner pore parameters of concrete with different BF contents; CT three-dimensional reconstruction technology was used to clarify the geometric characteristics of pores in the concrete. Through the comparative analysis of the transparent model, NMR, and CT test results, the mechanism of concrete performance deterioration was clarified. This is significant to further improve the gain effect of BF to concrete and optimize BF concrete’s mechanical properties in the future.

## 2. Specimen Preparation and Testing Procedure

### 2.1. Materials

#### 2.1.1. Basalt Fiber Concrete

In this work, we used Conch brand P.O 42.5 Portland cement, the chemical composition of which is shown in Table 1. The coarse aggregate was calcareous crushed stone with a maximum particle size of 10 mm; the fine aggregate was machine-made sand with a fineness modulus of 3.0 and particle size of 0–3 mm. We used short-cut basalt fiber (BF) 6 mm in length, produced by Shanxi Jin-Tou Basalt Development Co., Ltd., Datong, China. The physical and mechanical characteristics of the short-cut basalt fiber are shown in Table 2.

There were six different amounts of BF in the concrete (0 kg/m^3^, 1.5 kg/m^3^, 3 kg/m^3^, 4.5 kg/m^3^, 6 kg/m^3^, 7.5 kg/m^3^). The basic mix ratio of the concrete is shown in Table 3.

#### 2.1.2. Transparent Model

Quartz is a colorless and transparent silicate mineral. Quartz sand is produced by crushing quartz stone. High-purity quartz particles have excellent light transmittance and are wear-resistant and chemically stable. Quartz aggregate with the same gradation as concrete aggregate can be obtained by screening quartz sand with different particle sizes, which is a suitable material to simulate aggregate of concrete. However, the refractive index of quartz is quite different from that of air. When quartz particles are accumulated, light is irregularly refracted and reflected on the contact surface between air and quartz, resulting in the opaque state of quartz sand. If a liquid with the same refractive index as quartz is mixed with quartz aggregate, then light can pass through the mixture in a straight line, causing quartz aggregate to adopt a transparent state. Here, A-grade fused quartz sand was adopted, and seven specific particle sizes were screened: 0.1–0.15 mm, 0.15–0.3 mm, 0.3–0.6 mm, 0.6–1.18 mm, 1.18–2.36 mm, and 2.36–4.75 mm. The proportion of each type was as shown in Table 4.

Calcium bromide is an inorganic salt and a colorless crystal that is very soluble in water. The solution has excellent light transmittance and excellent temperature stability. The refractive index of CaBr_2_ solution can be adjusted in a relatively wide range. Therefore, CaBr_2_ solution was selected as the transparent medium. The parameters of calcium bromide are shown in Table 5.

### 2.2. Mechanical Test Method

In this study, the mixing, vibration, and pouring of concrete were all carried out indoors. The preparation method of basalt fiber concrete samples was as follows: stir the BF together with aggregate and cement for 30 s first; then, add water and admixtures and continue to stir for 120 s; after mixing, pour the concrete into a standard mold for vibration. The dimensions of the compressive and splitting tensile specimen were 100 × 100 × 100 mm, and the dimensions of the bending test block were 100 × 100 × 400 mm. The concrete specimens were cured at a temperature of 20 ± 2 °C and humidity > 90%, for 7 d, 14 d, and 28 d.

The mechanical test of BF concrete was carried out on a WES-1000B electro-hydraulic universal testing machine (accuracy: 0.01 kN, measuring range: 300 kN), which was designed by the Changchun Academy of Mechanical Sciences (Changchun, China). We followed the GB/T50081-2002 [19] standard for test methods regarding the basic mechanical performance of concrete. The loading rate for the compression test was 0.5 MPa/s, and that for the split tensile test and bending test was 0.05 MPa/s; the process of the test is shown in Figure 1.

### 2.3. Artificial Transparent Model Test

The distribution of BF in concrete can be simulated by using a transparent model, which consists of a transparent aggregate and liquid medium with the same refractive index. The schematic flow of the transparent model test is shown in Figure 2. Step 1: Screen quartz sand and adjust the grading of quartz sand to be consistent with that of the concrete aggregate. Step 2: Pour the quartz sand into a small mortar mixer and stir for 30 s, and then add the BF and stir for 1 min. Step 3: Prepare the CaBr_2_ solution by using an Abbe refractometer, and adjust the refractive index of the CaBr_2_ solution to 1.4585. Step 4: Pour the CaBr_2_ solution into the quartz sand; after 1 min, pour the mixture into a PMMA box (100 × 100 × 100 mm). Step 5: After the lumen of the lighting system is adjusted to the most appropriate value, the front of the model can be photographed with an industrial camera to obtain the characteristic distribution image of BF in the transparent aggregate.

### 2.4. Nuclear Magnetic Resonance

Nuclear magnetic resonance (NMR) is a powerful tool to study the inner structure of opaque materials. It can detect porosity, pore distribution, permeability, etc. In nuclear magnetic physics, the relaxation spectrum is suitable to quantify the rate of signal decay. The peak of the monitored signal decays exponentially; it represents a T_2_ relaxation spectrum. The T_2_ relaxation spectrum can reflect the characteristics of the liquid–solid interface, and the pore size determines the relaxation decay rate. Hence, the relaxation velocity directly reflects the pore structure information.

The equipment selected for this test was the ZYB-Ⅱ vacuum pressure saturation device and the MesoME23-040V rock microscopic pore structure analysis system, developed by Suzhou Niumay Analytical Instrument Co., Ltd., Suzhou, China; the magnetic field intensity of the instrument was 0.5 T.

The test was divided into five steps: (1) Sampling; (2) Vacuum water saturation; (3) Calibration; (4) Testing; (5) Data processing. Firstly, the concrete specimens with different BF contents were selected for coring and polishing, and then cylindrical specimens with 35 mm diameter and 50 mm height were produced. Because the NMR signal source is a hydrogen atom, it is necessary to fill the pores of the samples with water. After sufficient water saturation, we put the specimen into the instrument for testing. Then, the detected water signal was converted into pore information. In this way, pore information can be obtained. The flow of the nuclear magnetic resonance test is shown in Figure 3.

### 2.5. Computed Tomography

Computed tomography (CT) technology was used to obtain the internal images of samples. However, there will be noise interference during the scanning, so preprocessing is needed to remove noise points in images to ensure quality. Furthermore, image cutting was necessary as the data of the original CT image were quite large, which is not conducive to processing and observation. The flow mainly included median filtering, 3D reconstruction, threshold adjustment, pore extraction, etc. Then, the internal characteristics of the BF and concrete mix were obtained. Avizo software (Avizo 9.0, FEI Company, Hillsboro, OR, USA) was used in this process, as shown in Figure 4.

The HPC-225FB high-precision CT test system was used in this study; the working voltage of this system was 80 kV, the working current was 8 mA, the resolution was 10 μm, the number of scanning frames was 3 fps, and the pixel matrix was 1536 × 1930. The dimensions of the scanned BF concrete sample were consistent with the NMR sample (cylinder, 35 mm in diameter, 50 mm in height).

## 3. Mechanical Test Results

Figure 5 shows curves of the BF concrete’s strength at different ages. Figure 5a contains compressive strength curves, Figure 5b contains tensile strength curves, and Figure 5c contains flexural strength curves. The results showed that the compressive, tensile, and flexural strength increased first and then decreased along with the increase in BF content, shown as a para-curve with a downward opening. Table 6 presents the compressive, tensile, and flexural strength data of BF concrete with different BF contents.

Curing time was 7 d: the compressive strength reached the maximum value when the BF content was 3.0 kg/m^3^, approximately 23.12 MPa, which is higher than 0 kg/m^3^ BF content concrete by approximately 9.4%. However, when the BF content increased to 7.5 kg/m^3^, the compressive strength of the concrete decreased by 13.2% compared to 3.0 kg/m^3^ BF content. The tensile strength reached the maximum value when the BF content was also 3.0 kg/m^3^, at approximately 1.9 MPa. The tensile strength of 3.0 kg/m^3^ BF content increased by 20.2% compared to 0 kg/m^3^ BF content, but the tensile strength of 7.5 kg/m^3^ BF content decreased by 13.7% compared to 3.0 kg/m^3^ BF content concrete. The flexural strength reached 2.81 MPa when BF content was 3.0 kg/m^3^, 6.1% higher than 0 kg/m^3^ BF content. When BF content increased to 7.5 kg/m^3^, the flexural strength decreased by 6.4% compared to 3 kg/m^3^ BF content.

Curing time was 14 days: the compressive strength reached the maximum value when the BF content was 4.5 kg/m^3^, approximately 31 MPa, which was higher than 0 kg/m^3^ BF content concrete by around 9.9%. When the BF content increased to 7.5 kg/m^3^, the compressive strength of the concrete decreased by 10.3% compared to 4.5 kg/m^3^ BF content. The tensile strength reached the maximum value when the BF content was 3.0 kg/m^3^, around 2.42 MPa. The tensile strength of 3.0 kg/m^3^ BF content increased by 26.2% compared to 0 kg/m^3^ BF content, but the tensile strength of 7.5 kg/m^3^ BF content decreased by 22.2% compared to 3.0 kg/m^3^ BF content. The flexural strength reached 3.49 MPa when the BF content was 3.0 kg/m^3^, 8.1% higher than 0 kg/m^3^ BF content, and as the BF content increased to 7.5 kg/m^3^, the flexural strength decreased by 26.9% compared to 3 kg/m^3^ BF content.

Curing time was 28 days: the compressive strength reached the maximum value when the BF content was 3.0 kg/m^3^, approximately 41.2 MPa, which was higher than 0 kg/m^3^ BF content concrete by around 31.3%. As the BF content increased to 7.5 kg/m^3^, the compressive strength of the concrete decreased by 14.6% compared to 3.0 kg/m^3^ BF content. The tensile strength reached the maximum value when the BF content was also 3.0 kg/m^3^, approximately 2.62 MPa. The tensile strength of 3.0 kg/m^3^ BF content increased by 16.6% compared to 0 kg/m^3^ BF content, but the tensile strength of 7.5 kg/m^3^ BF content decreased by 13.2% compared to 3.0 kg/m^3^ BF content. The peak flexural strength was 3.9 MPa when the BF content was 3.0 kg/m^3^, and the flexural strength of 3.0 kg/m^3^ BF content increased by 13% compared to 0 kg/m^3^ BF content. As the BF content increased to 7.5 kg/m^3^, the flexural strength decreased by 6.3% compared to the peak flexural strength. 

It could be found that the best content range of BF in concrete is around 3.0–4.5 kg/m^3^. The mechanical properties of BF concrete will deteriorate when BF is used excessively; we hypothesize that the main reason for this phenomenon is the density of BF, which affects its dispersion form in a concrete aggregate, and then changes the microstructure of the concrete matrix. In the following, the test results of the transparent aggregate model are used to discuss the influence of content density on the morphological characteristics of BF concrete, and reveal the meso mechanism of the deterioration of BF concrete’s mechanical performance.

## 4. The Influence of BF Dosage on BF Dispersion

The model image and binary images of the BF distribution pattern in the transparent model with different BF contents are shown in Figure 6. A 50 mm × 50 mm section was cut from the BF transparent model, and the joint blocking method was used to process the test results. Based on the enhancement algorithm of quadratic function zooming to correct and enhance the original images, images with uniform brightness distribution were obtained. Then, the orientation, distribution, and contact point information of BF in the visual range were obtained, and we used a built-in extension module to explore the dispersion characteristics of basalt fiber in the concrete.

In order to identify BF’s scattered morphological characteristics in the visual field, the joint blocking method was used, as shown in Figure 7, and the image of the transparent model was divided into 4 × 4 blocks:

A_1_={P_1_,P_2_,P_5_,P_6_},A_2_={P_2_,P_3_,P_6_,P_7_},A_3_={P_3_,P_4_,P_7_,P_8_},A_4_={P_5_,P_6_,P_9_,P_10_},A_5_={P_6_,P_7_,P_10_,P_11_},A_6_={P_7_,P_8_,P_11_,P_12_},A_7_={P_9_,P_10_,P_13_,P_14_},A_8_={P_10_,P_11_,P_14_,P_15_},A_9_={P_11_,P_12_,P_15_,P_16_};
where A*_i_* is the image block number, *i* = 1, 2…, 9, and P*_i_* is the set of pixels in the block image.

The orientation and local color information of the visual field could be obtained by using the joint blocking method, and the HSV space was used for color feature extraction, as shown in Figure 7a.
(1)$$H_{l}=\frac{f_{i}}{N},l=1,2,3$$
where *H_l_* is three components in the HSV color system, *f_i_* is the number of pixels with color value *i* of each component, and *N* is the total number of pixels in the local block of the image.

The BF distribution position in the image is marked by the kernel function:(2)$$K(rsin\theta cos\varphi ,rsin\theta sin\varphi )=e^{-\chi^{2}(r,\psi )/\theta }=\prod_{j=1}^{n}\prod_{i=1}^{m}e^{-\chi_{i}{}^{2}\left(\gamma_{j}(i),\psi_{j}(i)\right)/\theta }$$
where *χ_j_*(*i*) and *ψ_j_*(*i*) are local color histogram vectors of the image, *i =* 1, 2, …, m, *θ* is the average value of the *χ*^2^ distance (BF spacing) of all blocks, and the calculation formula is as follows:(3)$$\chi^{2}(rsin\theta cos\varphi ,rsin\theta sin\varphi )=\frac{1}{2}\sum_{i=1}^{n}\frac{{\left(rsin\theta_{i}cos\varphi_{i}-rsin\theta_{i}sin\varphi_{i}\right)}^{2}}{rsin\theta_{i}cos\varphi_{i}+rsin\theta_{i}sin\varphi_{i}}$$
where *rsinθ_i_cosφ_i_* and *rsinθ_i_sinφ_i_* represent the BF position vector in the transparent model.

The color histogram kernel function *K_i_* can describe the imaging differences in the image; *χ*^2^ means can reflect the visual similarity of two kinds of color histograms, which provides an advantage for obtaining the BF’s morphological characteristics in the transparent model.

The dispersion coefficient *β* was introduced to evaluate the BF dispersion in the transparent model. The expression was as follows:(4)β=exp[−φ(rsinθcosφ)]
(5)$$\varphi \left(r,\theta ,\varphi \right)=\mu^{-1}\times {\left[\sum {\left(rsin\theta_{i}cos\varphi_{i}-\mu \right)}^{2}\times n^{-1}\right]}^{1/2}$$
where *μ* is the average number of fibers in the visual field, and *β* is the dispersion coefficient. Fibers dispersed well when *β* ∈ [0.5, 1], and fibers dispersed poorly when *β* ∈ [0, 0.5]. Figure 8 shows the dispersion coefficient of BF for different BF contents. The dispersion coefficient *β* of BF is negatively correlated with the BF content. When the BF content is 1.5 kg/m^3^, *β* is 0.92, and the effect of BF dispersion is the best.

Contribution index *C_f_* (*C*_0_ = 0, *f* = 0, 1.5, 3, 4.5, 6, 7.5) was defined to evaluate the gain effect of BF on strength. The expression is as follows:(6)$$C_{f}=\beta \cdot v^{-1}$$

The BF’s average contribution index with different BF contents is represented in Figure 8. The BF’s contribution index reached the largest value when the BF content was 3 kg/m^3^, *C*_3_ = 2.7. The contribution index was around 2.45 when the BF content was 6 kg/m^3^, and the contribution index was approximately 2.34 when the BF content was 7.5 kg/m^3^. It could be seen that C*_f_* decreased gradually with increasing BF content. It can be inferred that when the BF content exceeds the optimal value, the excess part of BF agglomerates and does not exert the gain effect.

Based on the above analysis, excessive BF content will cause the fiber agglomerates to form clusters. When the BF content is more than 3.0 kg/m^3^, there are a few fibers overlapping in the visual field. For BF content of 4.5 kg/m^3^, some fiber clusters come into contact and overlap, and when the BF content is more than 4.5 kg/m^3^, there is a dramatic increase in fiber cluster overlapping, the trend of dispersion coefficient is shown in Figure 9, and the mechanism of BF cluster overlapping is shown in Figure 10.

## 5. Effect of Fiber Dispersion on Pore Characteristics

The results of the transparent model test show that the BF overlapping and agglomeration lead to a reduction in its contribution index on concrete. However, the main reason for the deterioration in concrete performance has not been determined. In order to identify the main reason for the deterioration in BF’s concrete mechanical performance, nuclear magnetic resonance technology was used to detect the characteristics of the inner structure of BF concrete. Figure 11 presents the curves of the distribution of different-sized pores in BF concrete. The pores can be divided into three categories: gelation pores (*r* < 0.01 μm), capillary pores (0.01 μm < *r* < 5 μm), and connected pores (*r* > 5 μm). As BF concrete increases, the percentage of gelation pores decreases gradually, and there is a significant increase when the BF content increases from 3 kg/m^3^ to 4.5 kg/m^3^. This indicates that BF content of 3–4.5 kg/m^3^ is the sensitive range for the gelation pores, and 1.5–4.5 kg/m^3^ is the sensitive range for the capillary pores. When the BF content ≤ 3 kg/m^3^, the content of connected pores is low, but when BF content ≥ 3 kg/m^3^, the content of connected pores increases dramatically with the increase in BF content.

Because of the fiber clustering effect, there are only small gaps in between BF, which makes it challenging for the matrix to enter the fiber agglomerates, and there are crevices in the fiber agglomerates. These small gaps act as connectors to penetrate the primary pores in the concrete matrix, leading to changes in the pore structure in the concrete matrix. The total pore content increased with the increase in the BF content, and the total pore content significantly decreased with the increase in the curing time.

When the curing time reached 7 d, the main diameter pore component of the plain concrete sample was approximately 0.095%, but the main diameter pore component of concrete was approximately 0.126% when the BF content was 7.5 kg/m^3^; the pore component difference caused by the BF content reached 32.6%. When the curing time reached 14 d, the main diameter pore component of the plain concrete sample was approximately 0.091%, but the main diameter pore component of concrete was approximately 0.117% when the BF content was 7.5 kg/m^3^; the pore component difference caused by BF content reached 28.5%. When the curing time reached 28 d, the main diameter pore component of the plain concrete sample was approximately 0.093%, but the main diameter pore component of concrete was approximately 0.113% when BF content was 7.5 kg/m^3^; the pore component difference caused by BF content reached 21.5%. The reason for this is as follows: with the increase in the curing times, the hydration reaction produces CSH, AFT, AFM, CH, etc. These materials will fill the pores in the concrete matrix, leading the total pore ratio to decrease gradually.

In previous studies, connected pores (r > 5 μm) were defined as the key factor affecting the mechanical properties of concrete [20,21]. Hence, pores in concrete can be divided into two categories: harmful pores (connected pores) and harmless pores (gelation pores and capillary pores). Figure 12a shows the total porosity for different BF contents. It can be seen from Figure 12a that the porosity does not change significantly when the BF content ≤ 3 kg/m^3^. When BF content > 3 kg/m^3^, the porosity increases substantially with the BF content increase. Figure 12b shows the harmless pore ratio for different BF contents and Figure 12c shows the harmful pore ratio for different BF contents. It can be seen from Figure 12b,c that as the BF content increases, the harmless pore ratio decreases and the harmful pore ratio increases gradually. In particular, when BF content > 3 kg/m^3^, the percentage of harmful pores increases dramatically.

Reconstruction results of the pores in BF concrete are shown in Figure 13. The Feret diameter distribution in concrete was extracted. The mechanical properties of BF concrete were the best when the BF content was 3 kg/m^3^, and they were the worst when the BF content was 7.5 kg/m^3^. Therefore, BF content 3 kg/m^3^ and BF content 7.5 kg/m^3^ were compared and analyzed. Figure 13 shows the pore geometry and distribution characteristics of the Feret diameter. It can be seen from Figure 13b that the ratio of pores (Feret_max_ ≤ 1 μm) of the BF concrete sample (3.0 kg/m^3^) is higher than that of the BF concrete sample (7.5 kg/m^3^). For pores of Feret_max_ > 1 μm, the BF concrete sample (BF content 3.0 kg/m^3^) is significantly lower than that of the BF concrete sample (BF content 7.5 kg/m^3^). This indicates that the proportion of small pores decreases with the BF content increasing from 3.0 kg/m^3^ to 7.5 kg/m^3^; conversely, the proportion of large pores gradually increases with the BF content increasing from 3.0 kg/m^3^ to 7.5 kg/m^3^.

Table 7 shows the roundness and elongation corresponding to pore shape. When the BF content ≤ 3 kg/m^3^, pores with roundness of 0.604–1 account for 86.7%. When the BF content is 3 kg/m^3^, the pores with roundness of 0.310–0.502 account for 89.2%. When the BF content is more than 4.5 kg/m^3^, the pores with roundness of 0.310–0.502 account for 91.7%. This means that most of the pores are round when the BF content is low and most of the pores are irregularly shaped when the BF content is high, as shown in Figure 14.

## 6. Discussion

Compared with X-ray and CT scanning, the transparent model method is more suitable for observing the distribution of silicate fibers, but it still has certain shortcomings. (1) The density of quartz is slightly lower than that of actual concrete aggregates. If converted according to the ratio of BF weight/model volume, there is a slight deviation between the quantity of BF added in the transparent soil model and the quantity of BF in the concrete sample. (2) For cementitious materials, the viscosity of the cement has a certain effect on the distribution of aggregates and fibers. However, in the transparent model test, the viscosity of the CaBr_2_ has a certain gap with that of the cement slurry, which may cause the distribution of BF to deviate. Nonetheless, it does not affect the trend of the results. CT or X-ray scanning are only suitable for studying the distribution morphology of fibers (steel fibers, PVA fibers, etc.), which has a large gap with concrete in terms of density. Although the fluorescence tracer method has high accuracy, at present, the fluorescence tracking method is mainly used to detect the morphological form of synthetic fibers, which is easily labeled by a fluorescence tracer. For the silicate fiber, the effect of the fluorescence tracking method is still not ideal. Therefore, the detection method should be selected in a targeted manner for different types of fibers.

MIP is a powerful tool for studying pore parameters, such as pore radius, roar radius, and the pore–roar ratio of rock or concrete. This method has high accuracy, but the period for specimen preparation in the MIP test is too long, especially for the constant velocity mercury injection method. Typically, in order to shorten the test period, the specimen size must be as small as possible (around 1 cm^3^), so that the size is close to the maximum particle size of concrete aggregates. If such specimens are tested, the test results will not be general. NMR has the same accuracy as the MIP method to study the pore structure of BF concrete. The NMR method can detect samples with a relatively large size, and the NMR sample preparation is simpler; therefore, the NMR method was adopted in this study, combined with CT reconstruction, and the geometric characteristics of pores in concrete with different BF contents were compared and analyzed.

## 7. Conclusions

In this paper, the correlation and the internal mechanism between BF content, BF morphological characteristics, and the mechanical properties of concrete were explored through experiments. The following conclusions were drawn. (1)The mechanical test shows that BF concrete’s mechanical properties were the best when the BF content was around 3.0 kg/m^3^. However, the mechanical properties of concrete will deteriorate when BF is used excessively. At 28 d, the compressive, tensile, and flexural strengths of concrete with a BF content of 7.5 kg/m^3^ were lower by 14.6%, 13.2%, and 6.3%, respectively, than when the BF content was 3.0 kg/m^3^.(2)The transparent model test and joint blocking method were used to explain the morphological characteristics and dispersion mechanism of BF in the concrete matrix. The results show that when the BF content exceeds the critical value, the excess part of BF agglomerates and does not display a gain effect. The contribution index *C_f_* of BF was the largest when the BF content was 3 kg/m^3^, *C*_3_ = 2.7, but it was only 2.34 when the BF content was 7.5 kg/m^3^.(3)There is a strong correlation between BF’s morphological behavior and pore structure in the concrete matrix. The overlapping contact of BF clusters leads to the penetration of pores, which leads the macro-pore proportion to increase dramatically, and these macro-pores easily evolve into initial micro-cracks under loading. The increase in the macro-pore ratio is probably the main reason for the deterioration of concrete’s performance.

## 8. Future Work

The viscosity of cementitious materials was also a significant factor affecting BF’s morphological features. In this study, the viscosity of the CaBr_2_ solution was different from that of cement slurry, which may have caused the morphological properties of BF in the transparent model to differ from those in the concrete. Nonetheless, the trend of BF distribution in the transparent model with a change in BF content is the same as that in concrete. In future research, the transparent medium’s fluid viscosity should be also considered as an important factor.

## Figures and Tables

**Figure 1 materials-15-02788-f001:**
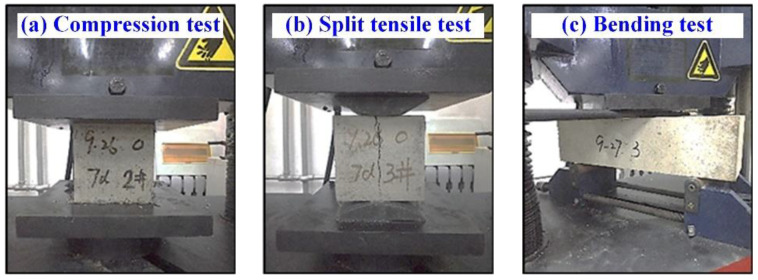
Basic mechanical properties test. (**a**) compression test; (**b**) split tensile test; (**c**) bending test.

**Figure 2 materials-15-02788-f002:**
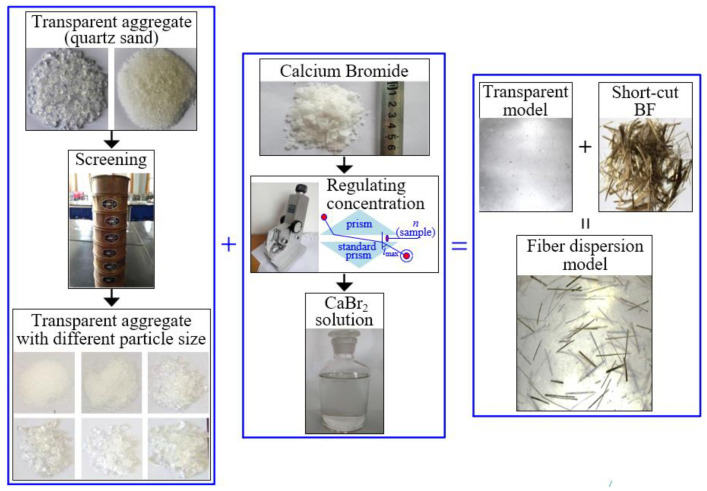
Schematic flow of transparent model test.

**Figure 3 materials-15-02788-f003:**
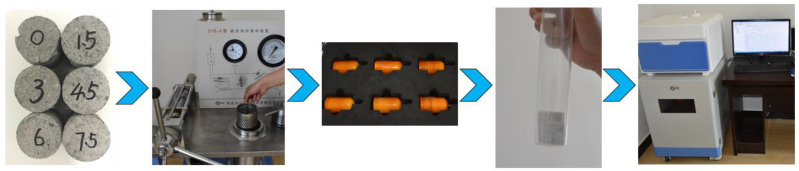
Experimental flow chart of the nuclear magnetic resonance experiment.

**Figure 4 materials-15-02788-f004:**
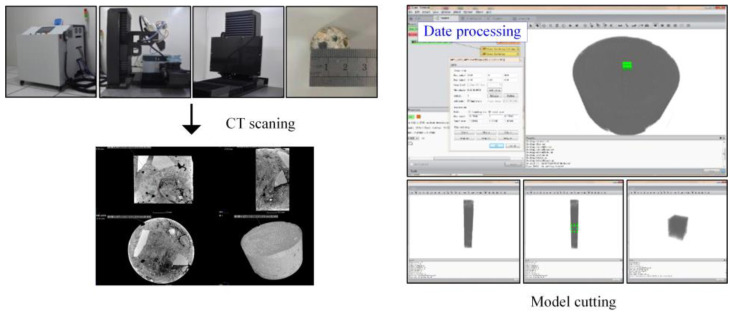
Test process of computed tomography.

**Figure 5 materials-15-02788-f005:**
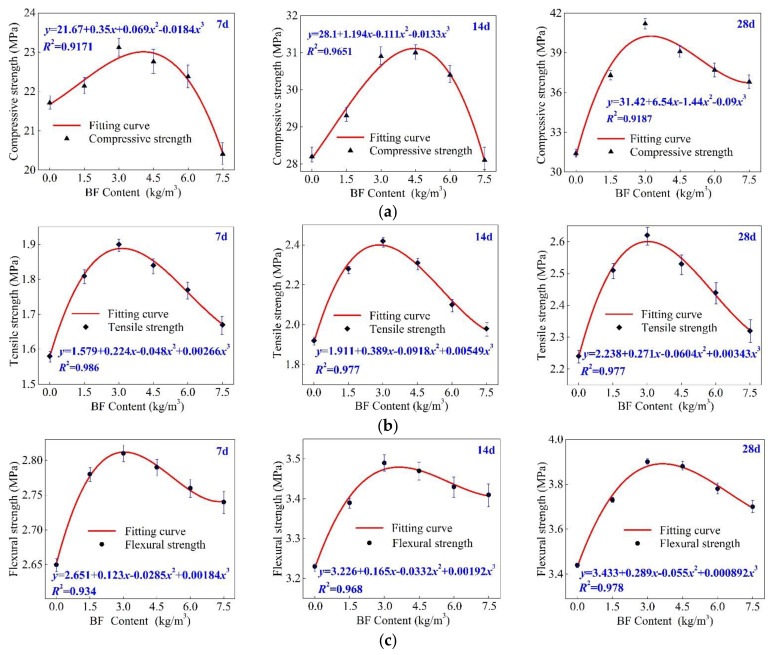
(**a**) Compressive strength. (**b**) Tensile strength. (**c**) Flexural strength (7 d, 14 d, and 28 d).

**Figure 6 materials-15-02788-f006:**
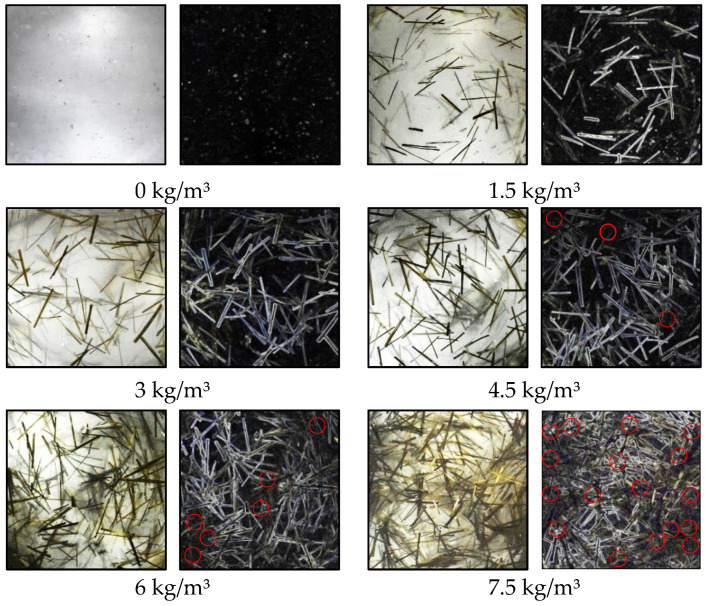
Binary images of the BF distribution pattern in transparent model with different BF dosages.

**Figure 7 materials-15-02788-f007:**
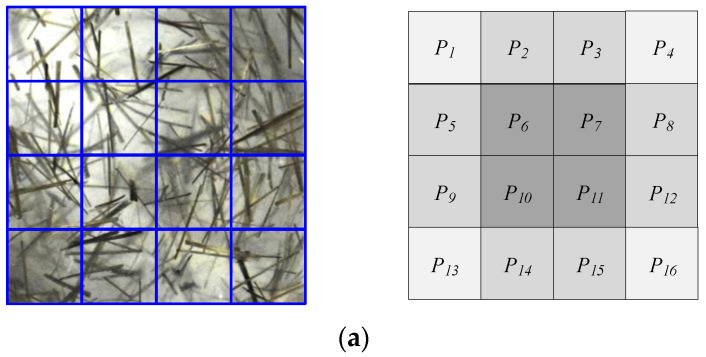
Image division and calibration. (**a**) Joint blocking; (**b**)Spherical coordinate.

**Figure 8 materials-15-02788-f008:**
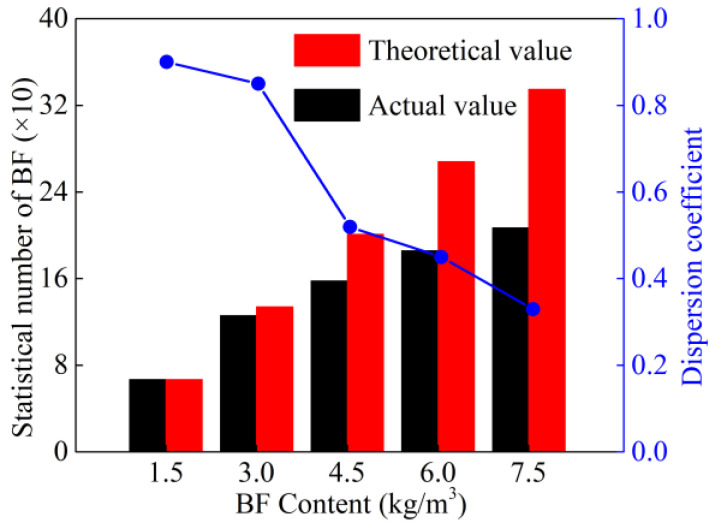
Dispersion coefficient of BF at different mixing ratios.

**Figure 9 materials-15-02788-f009:**
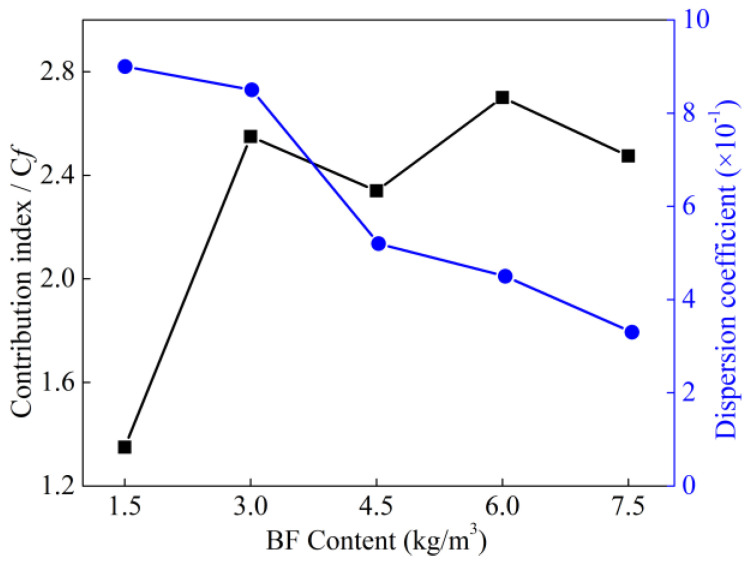
BF average contribution index under different BF contents.

**Figure 10 materials-15-02788-f010:**
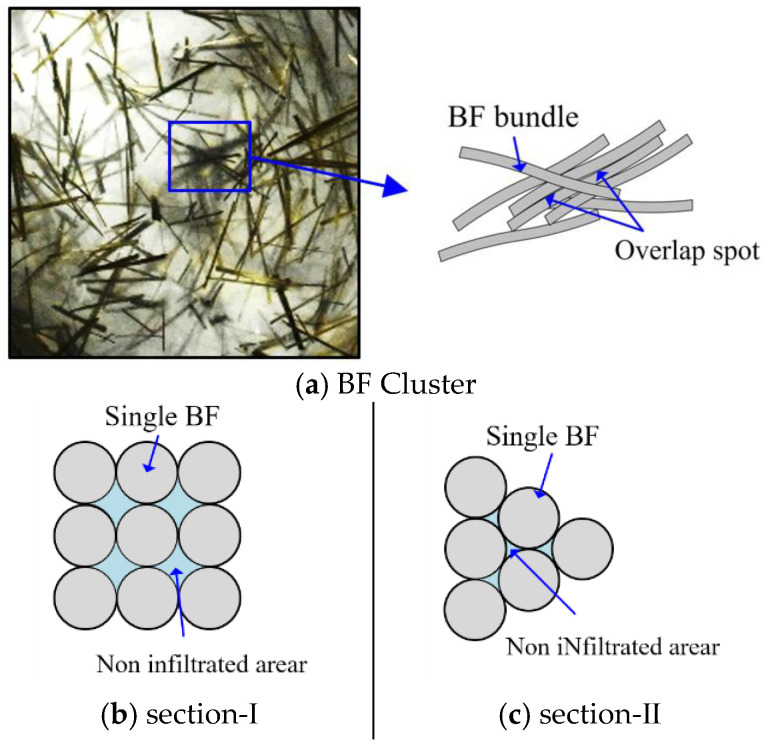
Cement-based unfilled area between fiber clusters.

**Figure 11 materials-15-02788-f011:**
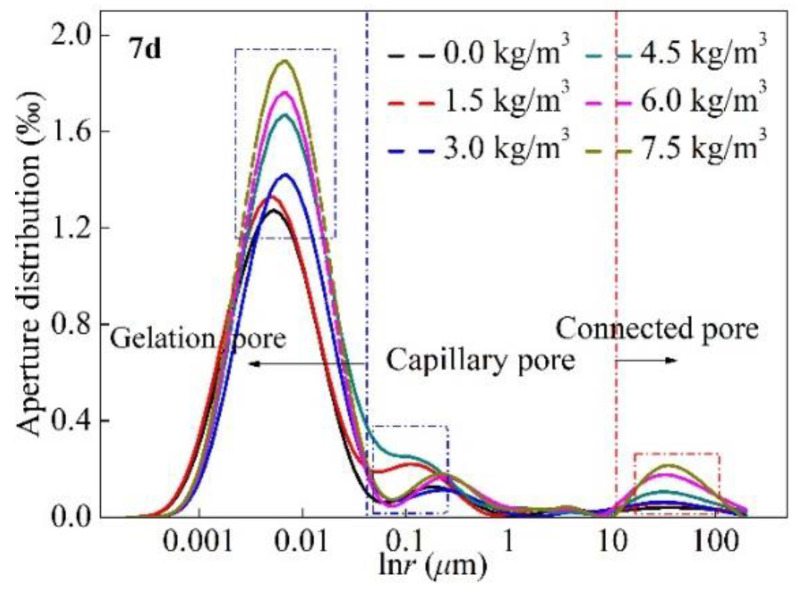
Distribution of different-sized pores in BF concrete (7 d, 14 d, and 28 d).

**Figure 12 materials-15-02788-f012:**
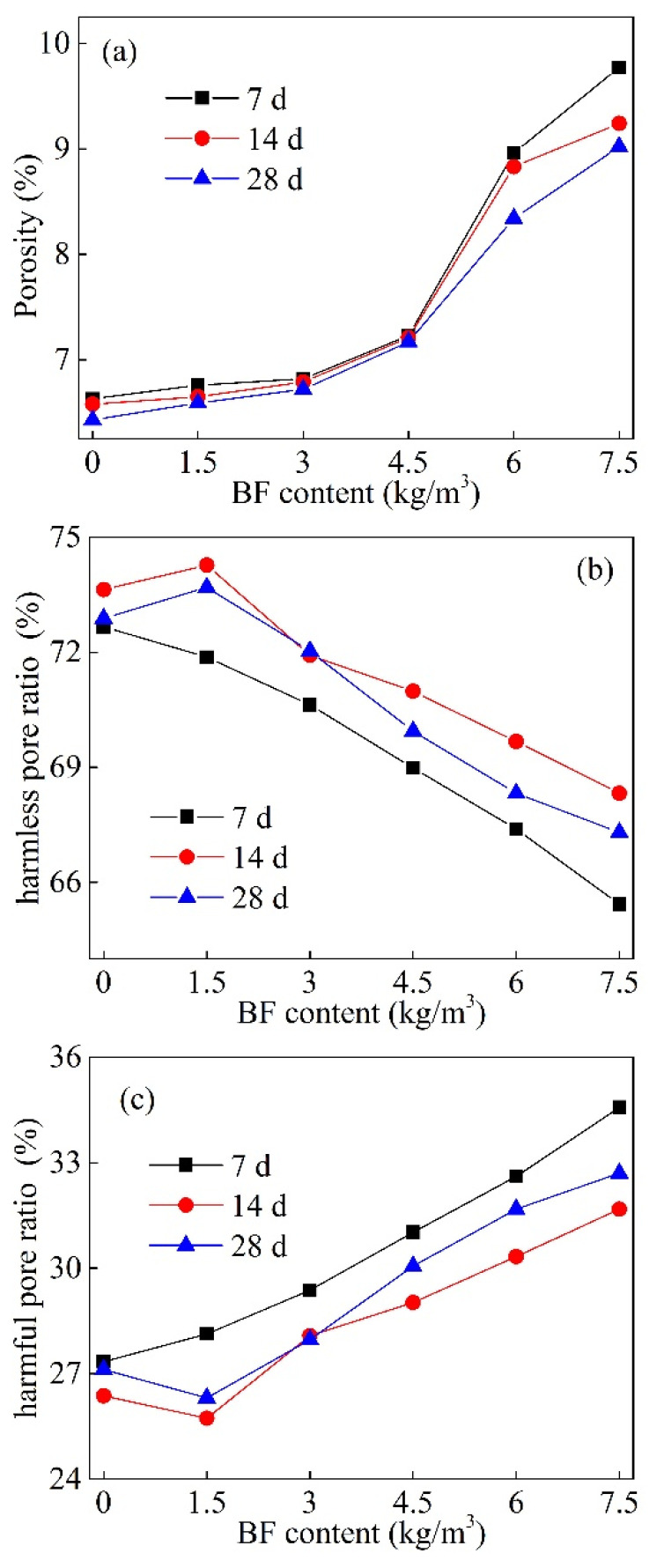
Pore ratio of different BF contents (**a**) porosity, namely, the ratio of total pores to the volume of BF concrete; (**b**) the ratio of harmless pores to the total pores; (**c**) the ratio of harmful pores to total pores.

**Figure 13 materials-15-02788-f013:**
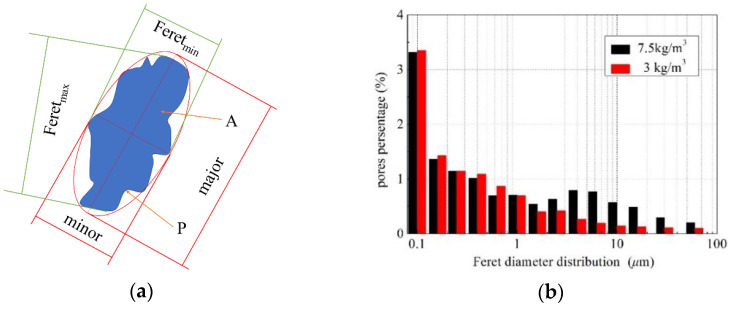
Schematic diagram of pore geometry and distribution diagram of Feret diameter. (**a**) pore geometry; (**b**) distribution of Feret diameter.

**Figure 14 materials-15-02788-f014:**
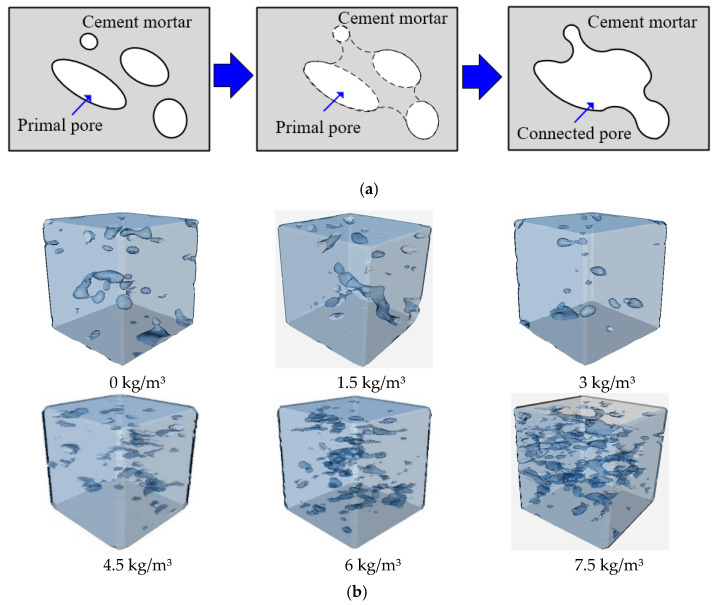
Three-dimensional reconstitution of internal pores of samples with different BF content ratios. (**a**) Primal pore penetration process; (**b**) Pore distribution.

**Table 1 materials-15-02788-t001:** Chemical composition of cement.

Composition	SiO_2_	Al_2_O_3_	Fe_2_O_3_	CaO	MgO	K_2_O	Na_2_O	SO_3_	LOI
Cement (%)	23.35	7.92	4.03	55.96	1.53	0.76	0.33	2.92	1.52

**Table 2 materials-15-02788-t002:** Physical and mechanical characteristics of short-cut basalt fiber.

Density/g·cm^–3^	Diameter/μm	Tensile Strength/GPa	Elastic Modulus/GPa	Fracture Elongation/%	Maximum Operating Temperature/°C
2.7–2.8	13	3–4	80–100	2.7–3.4	600–700

**Table 3 materials-15-02788-t003:** Basic concrete mix ratio.

Cement/kg	Sand/kg	Stone/kg	Accelerator/kg	Early Strength Agent/kg	Water/kg
440	880	880	17.61	2.2	260

**Table 4 materials-15-02788-t004:** Quartz sand grading.

Particle size range/mm	0.1–0.15	0.15–0.3	0.3–0.6	0.6–1.18	1.18–2.36	2.36–4.75	4.75–10	10–16
Percentage/%	10.7	4.0	10.3	17.5	12.6	31.9	5.3	7.0

**Table 5 materials-15-02788-t005:** Parameters of calcium bromide.

Density g/cm^3^	Boiling Point °C	Melting Point °C	Molecular Formula	Flash Point	Refractive Index
3.353	806	730	Br_2_Ca	806	1.583

**Table 6 materials-15-02788-t006:** Compressive, tensile, and flexural strength of BF concrete with different BF contents.

	BF Content (kg/m^3^)	0 kg/m^3^	1.5 kg/m^3^	3 kg/m^3^	4 kg/m^3^	6 kg/m^3^	7.5 kg/m^3^	
Strength Parameter								
Compressive strength/MPa		21.71	22.14	23.12	22.76	22.38	20.41	7 d
Tensile strength/MPa		1.58	1.81	1.9	1.84	1.77	1.67	
Flexural strength/MPa		2.65	2.78	2.81	2.79	2.76	2.64	
Compressive strength/MPa		28.2	29.3	30.9	31	30.4	28.1	14 d
Tensile strength/MPa		1.92	2.28	2.42	2.31	2.1	1.98	
Flexural strength/MPa		3.23	3.39	3.49	3.47	3.43	3.41	
Compressive strength/MPa		31.4	37.3	41.2	39.1	37.7	36.8	28 d
Increase ratio		100%	118%	131%	124%	120%	117%	
Tensile strength/MPa		2.24	2.51	2.62	2.53	2.44	2.32	
Increase ratio		100%	112%	116%	112%	108%	103%	
Flexural strength/MPa		3.44	3.73	3.90	3.88	3.78	3.70	
Increase ratio		100%	108%	113%	112%	109%	107%	

**Table 7 materials-15-02788-t007:** Roundness and elongation corresponding to pore shape.

Shape	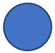		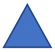		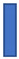	
Roundness	1.000	0.785	0.604	0.698	0.502	0.310
Elongation	1.000	1.000	1.000	2.000	4.000	8.000

## Data Availability

The data used to support the finding of this study are included within the article.

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
