# Peer review of "Study on Influence Mechanism of Short-Cut BF Dispersion Morphological Behavior on Concrete Properties Based on Meso Scale"

_materials, 2022, doi:10.3390/ma15082788_

Round 1

Reviewer 1 Report

Dear Authors,

This manuscipt is not prepared on satisfactory way. Please consider some of major comment on your manuscript:

In the chapter of 2.1. Specimens preparation should be improved procedure of preparation of specimens which is not defined on properly way. Chapter should be written on the way that experiment can be reproduced. In this chapter also include different shape of specimens for the measuring the Mechanical testing.

Table 1 needs to separate the physicochemical properties of BF from concrete. The description of concrete preparation does not indicate the physical characteristics of the concrete, but the content is an integral part of the preparation of the concrete sample that accompanies the experimental preparation of the sample.

In the chapter 2.2. Mechanical properties testing. Modify the description of chapter on the way that are given information are explained on systematically order. You don’t have any information about used equipment.

Chapter 2.3 Artificial transparent model is written on the way that is not understand what and how you use???

Chapter 2.5 Computed tomography, as all other chapter can be written with much more information. CT reconstructed model is poor and obtained data from analysis can be much more improved such as reconstruction of Pores size, pore size distribution, fibre distribution and its orientation in the specimens.

Reviewer 2 Report

The article is very interesting scientifically and valid, however it needs more revisions and clarifications that would make it publishable.

  • First, the English is poor, there are numerous syntax and grammatical errors in the text.
  • The introduction describes how the fibers are cost effective, what is the cost of 1 kg of basalt fibers?
  • It would be necessary to create a table with acronyms, currently it is extremely difficult to distinguish between acronyms in the article.
  • line 82 the word manteined is wrong, probably meant to say cured
  • As described by the authors, NMR detects hydrogen groups, but since the cement is hydrated as well as containing hydroxyl functional groups (as in portlandite), isn't there a risk of confusion between porosity and hydrated groups in the cement
  • For porosity measurement, isn't it sufficient to perform an analysis with mercury porosimeter to check the size and dimension of the pores created?
  • it is necessary to include a table with the mix design for each sample prepared in order to make more understandable to the reader the amount of each component used in the preparation of samples
  • Basalt fibers are silicates that generally have a good interaction with the cement paste and are compatible with the structure, however it is not clearly explained how the porosity at 28 days with the 7.5 Kg/m3 of fibers can be equal to the campioons at 14 days, as described by the authors the hydrated phases have more time to hydrate. Did the authors evaluate the effect of trapped air during sample preparation?
  • Was the viscosity of the slurry evaluated during its preparation? Because it was only mentioned in the conclusions.

Reviewer 3 Report

This is a well designed investigation of the mechanism of BF contribution on concrete mechanical properties. Please consider the following points for improving the quality of the manuscript.

  1. The language of the paper is poor. Extensive revision is necessary.
  2. Section 2: Concrete mix design is missing, as well as the type of cement used (chemical/mineralogical composition) and the applied curing regime.
  3. Section 2.2: Please cite the standard GB/T50081-2002.
  4. Section 2: Experimental details about the NMR and CT experiments are missing.
  5. Section 3: It is not necessary to show a table with the mechanical tests results (Table 2), since this information is included already in Fig. 5.
  6. In Fig. 13, (a) and (b) indicators are missing, although in line 290 "Fig. 13(b)" is mentioned.
  7. The relation of BF dispersion and material's viscosity was not investigated in this study, thus, the fourth conclusion should be deleted. This aspect can be included in the "Conclusion" section as a direction for future research, but not as a finding of the study.

Reviewer 4 Report

Kindly see the attached file. 

Reviewer 5 Report

The manuscript entitled ‘Study on influence mechanism of short-cut BF dispersion morphological behavior to concrete properties based on meso scale’ is in line with the Materials journal. However, it presents original research and the topic is up-to-date; the text has a serious flaws, including:

  • All article: English have to be corrected in this text.
  • Introduction: This part must be re-written, the references in the first part seem to be put into the text quite accidentally. They do not refer to specific problems and are connected only for the general sentence (lines 30-35). Please refer specific research given in this article not general statements. Second problem is that references are given incorrectly, for example line 36, 45 and many others. The given reference is to one author, but the number referred a lot of position also written by other research teams. Next flaws is lack of reference in the places where are needed, for example, with precents in line 46 (but also other places). In this form, introduction cannot be accepted.
  • Introduction: Please stress the novelty aspects in the presented research (last paragraph).
  • Chapter 2: please add manufacturer for the used equipment (all sub-chapters).
  • Table 1. It should be divided into 2 tables one connected with fibre characteristic and second with mix composition. The elements of mix composition required more detailed description (in the table or in the text), including kind of used aggregates, sizes, etc. (and many other information).
  • Chapter 2.3: The applied model requires to be justified according the relevance to ‘real’ material; potential weakness of the model should be also included in the text.
  • Chapter 2.4: please add more details for porosity research (method description).
  • Chapter 2.5: information about the dimensions of samples and their preparation for each research are required (also chapter 2.4.).
  • Table 2: lack of unit.
  • Chapter 5. More detailed information about measurement method are required, it should be supplemented in part 2.
  • Discussion: Please add proper discussion (not a short summary of the results obtained). Discussion should include comparison of obtained results with the state-of-the-art, especially up-to-date literature.

Round 2

Reviewer 1 Report

-

Author Response

Dear reviewer

         Thank you for your comments and contributions to the progress of the manuscript !

Best regards,

Zilu

Reviewer 2 Report

The authors addressed all the comment, and it is acceptable

Author Response

(The authors gave the same response as above.)

Reviewer 3 Report

The manuscript was improved as for reviewers comments. Please consider the following points.

  1. The chemical and mineralogical composition of the cement used are missing.
  2. Figs. 1, 3 and 4 are not necessary.
  3. Why the authors show the curve fitting equations in Fig. 5?  

Author Response

Dear reviewer

         Thank you for your comments and contributions to the progress of the manuscript ! please see the attachment. 

Best regards,

Zilu

Reviewer 4 Report

No further comments. 

Author Response

(The authors gave the same response as above.)

Reviewer 5 Report

The manuscript entitled ‘Study on influence mechanism of short-cut BF dispersion morphological behavior to concrete properties based on meso scale’ was partly improved, however the changes are not marked in the text. The article still requires some changes:

  • Introduction: Please define the lterature crearly and add information about novelty aspects in comparison with analyzed literature; please justify why this research are necessary.
  • Sub-chapter 2.1.2. Transparent model needs claryfication. There is description the quartz as a mineral mixed with description of quartz sand. What exacly was applied? 
  • Chapter 2: Please describe model limitations. 
  • Discussion: In this part is still lack of comparison of obtained results with the state-of-the-art, especially up-to-date literature.

Author Response

Dear reviewer

Thank you for your comments and contributions to the progress of the manuscript !  Enclosed is the revised version of the paper as well as a separate file explaining the changes we have made in response to the comments. 

Best regards,

Zi-Lu

YeLi
